# Comprehensive Evaluation of Ecological-Economic Value of Guangxi Based on Land Consolidation

Lili Zhang [1,2,3,4], Baoqing Hu [2,3,4,*], Ze Zhang [5], Gaodou Liang [1] and Simin Huang [2,3,4]

1    School of Natural Resources and Surveying, Nanning Normal University, Nanning 530001, China; zhanglili1999@163.com (L.Z.)
2    Key Laboratory of Environment Change and Resources Use in Beibu Gulf, Ministry of Education, Nanning Normal University, Nanning 530001, China
3    Guangxi Key Laboratory of Earth Surface Processes and Intelligent Simulation, Nanning Normal University, Nanning 530001, China
4    School of Geography and Oceanography, Nanning Normal University, Nanning 530001, China
5    Department of Geographical Sciences, Beijing Normal University, Beijing 100000, China
*    Correspondence: hbq1230@nnnu.edu.cn

**Abstract:** The "Two Mountains" concept of "green water and green mountains are gold and silver mountains" plays an important value-oriented role in the ecological transformation of land consolidation. In this study, Guangxi was divided into five consolidation zones in combination with relevant policies, and the evolution characteristics and change intensity of ecological-economic values before and after the three phases of land consolidation in Guangxi and each consolidation zone in 2010, 2015 and 2020 were explored by bivariate spatial autocorrelation, standard deviation ellipse, and linear regression equation. Finally, the ecological-economic values of each consolidation area, which were obtained separately, were standardized by z-score, and the standardized results were matched by dividing quadrants for analysis. The ecological-economic value matching states of each consolidation area are ecological-economic value coordinated development type (central karst basin area of Guangxi), ecological value imbalance type (southeast plain area and coastal hilly plain area of Guangxi), economic value imbalance type (northwest mountain area of Guangxi) and ecological-economic value low imbalance type (northeast hilly mountain area of Guangxi). The study aims to provide a theoretical basis for the planning and differentiated management of land consolidation in Guangxi and promote the ecological-economic value transformation of the region.

**Keywords:** land consolidation; ecological-economic values; comprehensive evaluation; Guangxi

## 1. Introduction

The 20th Party Congress report pointed out that "we must firmly establish and practice the concept of green water and green mountains is the silver mountain of gold, standing at the height of the harmonious coexistence of man and nature planning development", and show new action in the construction of ecological civilization. The "Two Mountains" theory was created and developed to coordinate economic development and ecological protection, and has become an important part of Xi Jinping's thoughts on ecological civilization [1,2]. The "green mountain" is the ecological benefit generated by the ecosystem, while the "golden mountain" is the socio-economic benefit brought by the socio-economic system based on the ecosystem [3,4]. The "Two Mountains" theory is a scientific assertion of the dialectical relationship between ecological and socio-economic benefits [5,6]. Therefore, with the deepening of the "Two Mountains" theory, it is important to explore the characteristics and laws of ecological-economic wealth change for the sustainable green development of the region. The Ministry of Land and Resources released the "National Land Development and Finishing Plan (2001–2010)" in 2003, which pointed out that "land development and finishing is an important way to replenish arable land, achieve a balance

of arable land, improve production conditions and ecological environment, and increase land production capacity" [7]. At present, the main form of land development and finishing in China is land consolidation [8,9]. Land consolidation activities can often directly change the land use structure and constitute the ecological environment within the planning area, such as soil surface [10], vegetation [11], hydrology [12], and other basic ecosystem elements [13,14], thus directly or indirectly affecting the ecological environment changes in the whole planning area [15–17]. The economic and social benefits generated by human activities depend on the existence of the ecosystem, and the economic and social benefits will eventually feed back to the ecological environment and affect the increase or decrease in ecological benefits [18,19]. Therefore, the ecological benefits, economic benefits and social benefits are interdependent and affect each other. In 2005, Comrade Xi Jinping put forward the famous green development concept of "green water and green mountains are golden mountains" in Anji, Zhejiang Province, to address the issues of protection and development [20]. This plays an important role in guiding the value of ecological transformation of land management [21]. While promoting regional economic and social benefits, land consolidation can also have multiple impacts on the ecological benefits generated by natural ecosystems [22,23]. These effects require ongoing attention and moderation [15,24].

Previous studies often focused only on ecology and economics and explored regional sustainable development at the national [25], regional [20], provincial [1], municipal [26], and district scales [27,28] through methods such as the value of ecosystem services [29], the energy value approach [20], coupled coordination models [1], and spatial econometrics [26]. Other studies focus only on the scale and effect evaluation of land consolidation [30–32], plot layout and tenure adjustment [33,34], and land restoration technology and engineering [35–37]. Since the 18th Party Congress, with the deepening of ecological civilization ideology, the evaluation of ecological effects of land consolidation has become one of the hot spots of relevant research, and the ecosystem service value method is the most commonly used quantitative evaluation method [23,38,39]. Due to the heterogeneity of the natural environment, land consolidation activities have different impacts on the value of ecosystem services at different scales. In recent years, more studies have focused on the changes in ecological service value gains and losses before and after land consolidation, mostly focusing on provincial [40,41], municipal [42], county [43], and watershed scales [44]. In addition, the evaluation of land consolidation projects also generally focuses on the evaluation of land use intensification during and after project implementation and the evaluation of economic benefits after finishing [45–47]. The social and ecological benefits brought by project implementation are mostly expressed in qualitative terms, with little quantitative analysis and evaluation. The spatial and temporal evolution of ecological and economic values of land consolidation areas based on the perspective of land consolidation is not explored. From the perspective of ecological environment improvement and land use benefit enhancement, land consolidation zoning is zoned according to the differences and similarities in the ecological service function enhancement needs of different regions, so that effective and reasonable engineering measures can be configured. Land consolidation zoning oriented to the enhancement of ecosystem service functions is a prerequisite for the ecological transformation of land consolidation [44]. For this reason, in the context of the "Two Mountains" theory, the research framework of land consolidation is broadened to include the synergistic development of socio-economic wealth and ecological wealth. For this reason, in the context of the "Two Mountains" theory, the research framework of land consolidation is broadened to include the synergistic development of socio-economic wealth and ecological wealth, uncovering the laws of ecological and economic value changes in land consolidation areas of different landscapes and scales, and clarifying the development status of each land consolidation area under the matching of ecological and economic values, instead of exploring the impact of land consolidation on the ecosystem from a single value.

Guangxi is one of the core areas of karst desertification management in China, with a fragile ecological environment, large regional variability in land use and ecosystem services, and serious degradation of land ecosystems in the region [48,49]. During the "Twelfth Five-Year Plan" and "Thirteenth Five-Year Plan" periods, the Guangxi government

issued documents such as the "Guangxi Land Consolidation Plan (2010–2015)" and the "Guangxi Zhuang Autonomous Region Land Consolidation Plan (2015–2020)" in order to vigorously promote the treatment and restoration of abandoned and damaged land and promote the sustainable use of land resources. The effective implementation of various land consolidation policies has strongly promoted the regional land consolidation work. The region's land consolidation work achieved great results, including effectively increasing the area of arable land, improving the quality of arable land, ensuring food security, and improving rural production and living conditions. Therefore, it is necessary to do a good job in the process of land consolidation planning and project construction to evaluate and study the impact of ecological and socio-economic benefits on Guangxi.

In view of this, this study divides Guangxi into five major consolidation zones in conjunction with relevant policies and explores the characteristics and intensity of changes in ecological-economic value evolution before and after the three phases of land consolidation in Guangxi and different consolidation zones in 2010, 2015 and 2020 by means of bivariate spatial autocorrelation, standard deviation ellipse, and linear regression equation. Finally, we standardized the z-score of the ecological-economic value of each consolidation area obtained separately, and the standardized results were matched by dividing the quadrants for analysis. The study aims to provide a theoretical basis for the planning and differentiated management of land consolidation in Guangxi and promote the value transformation of ecological-economic values in the region.

## 2. Study Area and Data Sources

### 2.1. Study Area

With an administrative area of 237,600 square kilometers, Guangxi (104°29′~112°04′ E, 20°54′~26°23′ N) has 14 prefecture-level cities and jurisdiction over the Beibu Gulf sea area of about 40,000 square kilometers (Figure 1). For a long time, unreasonable exploitation and rough use resulted in different forms of human-land conflicts such as environmental pollution, rock desertification, soil erosion, and soil heavy metal pollution, while measures for the protection and management of regional natural resources have lagged behind. Combined with the Land Consolidation Plan of the Guangxi Zhuang Autonomous Region and the differences in landform types and natural climate from north to south in the province, Guangxi has been divided into five improvement areas according to the naming method of "area + typical landform type" and different improvement and construction focus directions based on the agricultural industry pattern, integrated consideration of urbanization construction patterns and ecological protection functions, the strategic pattern of land development in Guangxi's main functional area, consideration of administrative boundaries, and comprehensive natural, economic, social, historical, and other factors. These areas of Guangxi include the northeast hilly mountain area, the southeast plain area, the central karst basin area, the coastal hilly plain area, and the northwest mountain area. The northeast hilly mountain area of Guangxi mainly includes Guilin city and Hezhou city. The southeast plain area of Guangxi includes Yulin city, Guigang city, and Wuzhou city. The central karst basin area of Guangxi includes Liuzhou city and Laibin city. The coastal hilly plain area includes Nanning, Beihai, Fangchenggang, and Qinzhou. The northwest mountainous area of Guangxi includes Baise City and Hechi City.

### 2.2. Data Sources

The land use data on the study area from 2010–2020 were obtained from the Resource and Environment Science Data Center of CAS with a spatial resolution of 30 m × 30 m. The digital elevation model (DEM) data were obtained from the geospatial data cloud, the socio-economic data were obtained from the Guangxi Statistical Yearbook, and the grain production and grain price statistics were obtained from the China Agricultural Products Price Survey Yearbook.

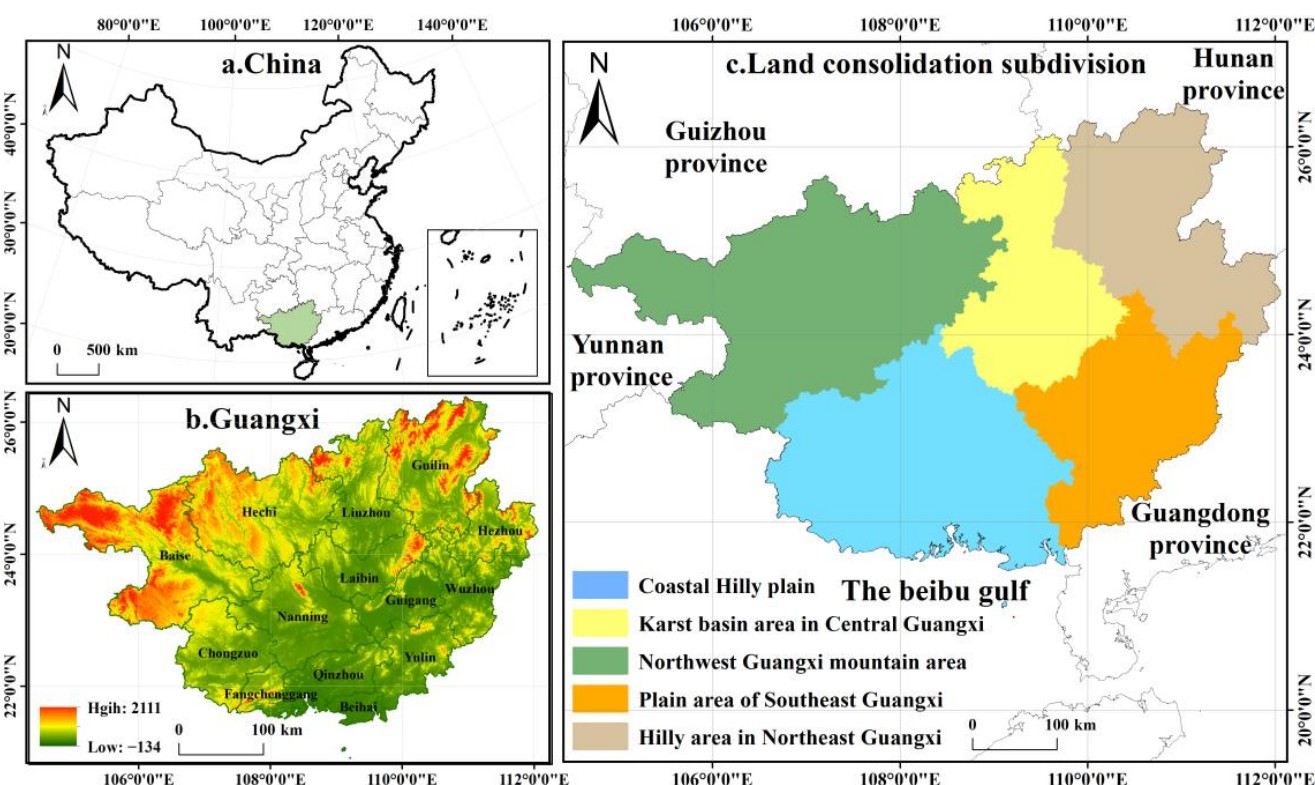

**Figure 1.** Guangxi land consolidation zoning map.

## 3. Research Methods

### 3.1. Land Consolidation and Ecological-Economic Values

First, this study considers that the value of goods and services provided by regional ecosystems constitutes ecological wealth. Ecological services consist of four types: supply services, regulation services, cultural services, and support services [50]. Previously, in the context of traditional economics, ecosystem services and products were considered to be valuable assets given to humans by nature for free and without any value [51]. This definition denies that benefits from ecosystem services and products in the promotion of socioeconomic development should be included in the framework of wealth. In 1997, Costanza et al. (1997, 2017) [52,53] published their results in *Nature*, suggesting that "ecosystem services and the stock of natural capital that produces them represent a fraction of the total economic value of the planet" and quantifying the value of ecosystem services in the context of the global scale. The Millennium Ecosystem Assessment Framework was developed by the United Nations in 2005. This has not only made the study of ecosystem service types, valuation methods, and principles gradually become one of the academic hotspots in geography, ecology, and ecological economics, but also promoted the move from knowledge production to public decision-making to explore ecosystem services [54]. Therefore, using the results of ecosystem service valuation as the quantitative value of ecological value is consistent with the direction of human understanding and development, and has a scientific basis.

Second, this study considers that the value of goods and services provided by the human socioeconomic system constitutes material wealth, i.e., the national economic accounting index GDP characterizes economic value. In the 1870s Adam Smith, the famous British economist and pioneer of classical economics, defined national wealth as all the conveniences and necessities of life that the nation consumes every year [55]. In the 1970s, the neoclassical school of economics was influenced by the utility-value theory to expand the concept of wealth. The SNA system it constructed has become the international standard for national wealth accounting systems [55]. GDP, on the other hand, is the most important aggregate indicator in the national economic accounting system. Although the current

GDP does not fully reflect the true wealth level of the region and the accounting system still has certain defects, it includes the value of agricultural products, industrial products, and other socio-economic services, which can more completely encompass the value of various products and services supplied by human economic and social development.

Land consolidation and eco-economic values are closely related and interact with each other (Figure 2). Under the current concept of ecological civilization construction, the methodological path of land consolidation optimization zoning is carried out from the perspective of ecological-economic value. The first step is to understand the regional environmental background information, calculate the ecological-economic value, quantitatively analyze and evaluate the ecological and economic value status, and clarify their spatial patterns and differentiation characteristics. Second, the spatial association and dependence characteristics of both are identified by bivariate spatial autocorrelation. Again, the location of the center of gravity of its area and its direction of spatial expansion are inscribed based on the standard deviation ellipse method. Finally, the trends of the two values are defined based on linear regression equations, and the matching status of ecological value and economic value in the five major consolidation areas is analyzed and identified.

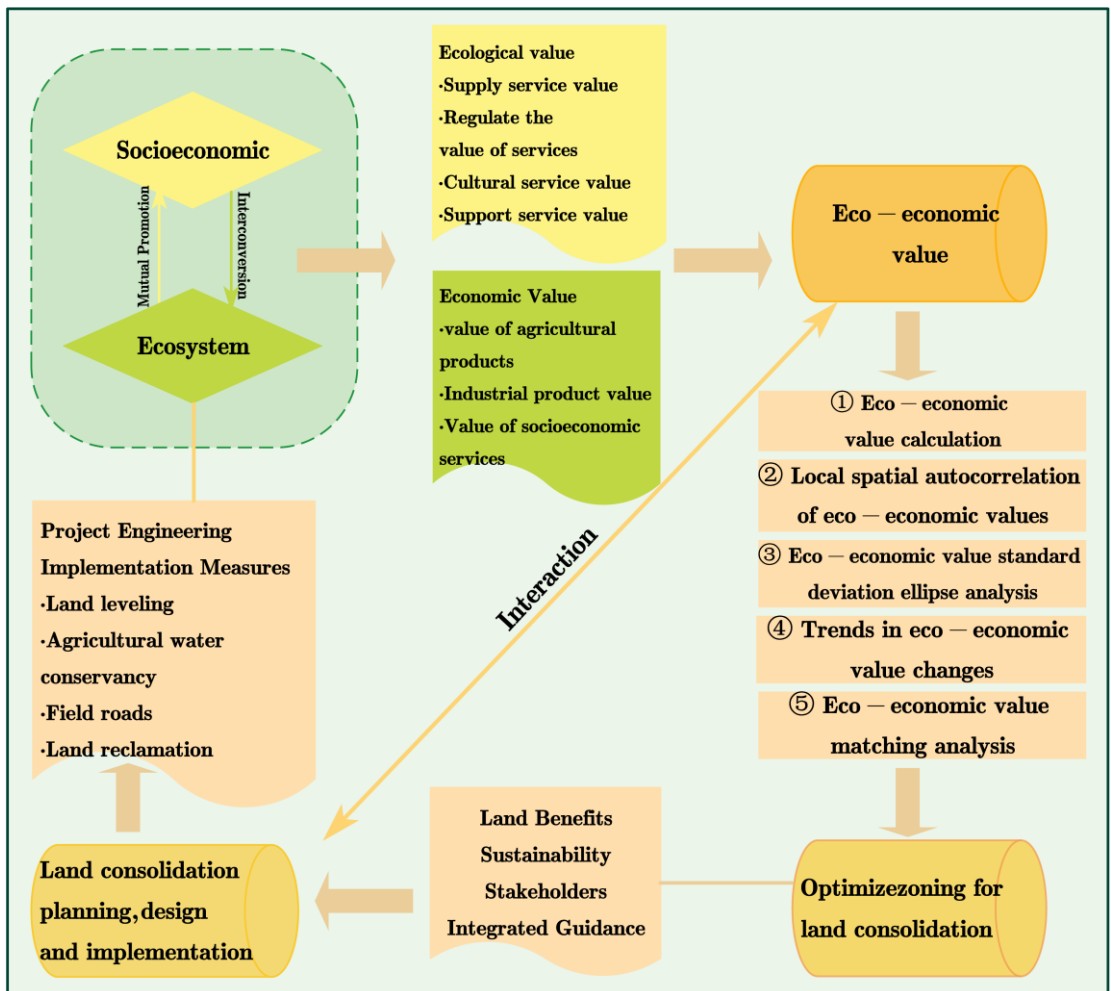

**Figure 2.** A framework for linking land reclamation and ecological-economic values.

### 3.2. Ecological-Economic Value Calculation

3.2.1. Ecological Value Calculation

To estimate the value of ecosystem services in the Chinese region, Xie et al. [56–58] conducted a questionnaire survey of several Chinese experts in the field of ecology, revised Costanza's results, and developed a Chinese ecosystem service value equivalent factor

table. According to the natural economic situation of different provinces, the correction coefficient is proposed, and the correction coefficient of Guangxi is 0.98. The ecosystem service value equivalent factor coefficient is based on the relative contribution of ecosystem services that can be generated by different ecosystems and has a value of about 1/7 of the average market value of grain yields in the study area. With reference to relevant studies and agricultural production in the study area in 2020 [59,60], an equivalent factor for the value of ecosystem services in the study area was determined to be 2034.87 Yuan/ha. This value was used as a benchmark to determine the ecosystem service value coefficients for each land use type in the study area, as shown in Table 1. Finally, the value of ecosystem services was calculated and the relevant formula was as follows.

$$VC = \frac{1}{7}\sum_{i=1}^{n} \frac{m_i \times p_i \times q_i}{M} \tag{1}$$

where *VC* denotes the economic value of food production services provided by 1 ha of agroecosystem, $m_i$, $p_i$ and $q_i$ represent the sown area of a food crop, the average market value of food yields in the study area, and the yield per hectare, respectively.

$$ESV = \sum_{j=1}^{m} \sum_{i=1}^{n} (LUC_i \times VC_{ij}) \tag{2}$$

where *ESV* indicates the total annual value of ecosystem services, $LUC_i$ is the area of the ith land use type, and $VC_{ij}$ indicates that the area of the ith land use type is equivalent to its jth ecosystem service.

**Table 1.** The monetary equivalent of the ecological services that a unit area ecosystem provides.

| Primary Type | Secondary Type | Farmland | Forestland | Grassland | Water Area | Construction Land | Unused Land |
|---|---|---|---|---|---|---|---|
| Supply services | Food supply | 1995.96 | 658.67 | 858.26 | 888.20 | 0.00 | 39.92 |
| | Raw material supply | 778.42 | 5947.96 | 718.55 | 588.81 | 0.00 | 79.84 |
| Regulating services | Gas regulation | 1437.09 | 8622.55 | 2993.94 | 2914.10 | 0.00 | 119.76 |
| | Climate regulation | 1936.08 | 8123.56 | 3113.70 | 15,578.46 | −958.06 | 259.47 |
| | Water Harvesting | 1536.89 | 8163.47 | 3033.86 | 32,144.93 | −2175.60 | 139.72 |
| | Waste care | 2774.38 | 3433.05 | 2634.67 | 29,190.91 | 0.00 | 518.95 |
| Support services | Soil conservation | 2934.06 | 8023.76 | 4470.95 | 2395.15 | 0.00 | 339.31 |
| | Maintaining biodiversity | 2035.88 | 9001.78 | 3732.44 | 7105.62 | 0.00 | 798.38 |
| Cultural services | Provide aesthetic value | 339.31 | 4151.60 | 1736.48 | 9111.56 | 0.00 | 479.03 |
| | Total | 1995.96 | 658.67 | 858.26 | 888.20 | 0.00 | 39.92 |

### 3.2.2. Economic Value Calculation

The eco-efficiency coefficients for each region were obtained by the equivalence coefficient method outlined in Section 3.2.1. The economic efficiency coefficients of farmland, forestland, grassland, and water areas were obtained from the statistics of agricultural output value, forestry output value, pasture output value, and fishery output value in the study area. The economic benefits of the construction land are calculated based on the total output value of the secondary and tertiary industries (Table 2).

**Table 2.** Economic eco-efficiency coefficients per unit area ($10^4$ Yuan/km$^2$).

| Efficiency Factor | Farmland | Forestland | Grassland | Water Area | Construction Land | Unused Land |
|---|---|---|---|---|---|---|
| Economic efficiency coefficient | 643.43 | 28.19 | 690.04 | 1375.63 | 30,317.5 | 0 |
| Eco-efficiency coefficient | 157.68 | 561.26 | 232.93 | 999.18 | −31.34 | 27.74 |

*3.3. Bivariate Spatial Autocorrelation*

The bivariate spatial autocorrelation can visually and clearly reflect the clustering phenomenon of the study unit as a whole and locally. The method is highly applicable and effective in describing the spatial association and dependence characteristics of two geographic elements by judging whether their aggregation characteristics exist in space and reflecting whether the mean value between a variable and another variable in its proximity is high-high, low-low, high-low, or low-high. Its calculation formula is as follows [60].

$$I_{ki}^i = \frac{X_k^i - \overline{x}_k}{\sigma_k} \sum_{j=1}^{n} \frac{w_{ij}\left(X_l^j - \overline{x}_l\right)}{\sigma_l} \tag{3}$$

where: $I_k^i$ denotes the spatial autocorrelation coefficient; $w_{ij}$ is the spatial connectivity matrix of regions $I$ and $j$; $X_k^i$ is the value of attribute $i$ of region $k$; $X_l^i$ is the value of attribute $j$ of region $l$; $\overline{x}_k$ and $\overline{x}_j$ are the means of attributes $k$ and $l$; $\sigma_k$ and $\sigma_l$ are the variances of attributes $k$ and $l$.

*3.4. Standard Deviation Ellipse*

The standard deviation ellipse is used to analyze the geographical distribution of ecological-economic values and to carve out the location of the center of gravity of its area and its spatial expansion direction. Its elements consist of the center, azimuth, long axis, and short axis, and the formula is as follows [28].

$$\overline{X}_w = \frac{\sum\limits_{i=1}^{n} w_i x_i}{\sum\limits_{i=1}^{n} w_i}, \overline{Y}_w = \frac{\sum\limits_{i=1}^{n} w_i y_i}{\sum\limits_{i=1}^{n} w_i} \tag{4}$$

$$\theta = \arctan\left[\left(\sum_{i=1}^{n} x_i'^2 - \sum_{i=1}^{n} y_i'^2\right) + \sqrt{\left(\sum_{i=1}^{n} x_i'^2 - \sum_{i=1}^{n} y_i'^2\right)^2 + 4\left(\sum_{i=1}^{n} x_i' y_i'\right)^2}\right] / 2\sum_{i=1}^{n} x_i' y_i' \tag{5}$$

$$\delta_x = \sqrt{\sum_{i=1}^{n} \left(x_i' \cos\theta - y_i' \sin\theta\right)^2 / n}, \delta_y = \sqrt{\sum_{i=1}^{n} \left(x_i' \sin\theta - y_i' \cos\theta\right)^2 / n} \tag{6}$$

where: $(X_w, Y_w)$ is the weighted mean center; $(X_i, Y_i)$ are the coordinates of the geometric center of the county; $W_i$ is the weight; $\theta$ is the elliptical azimuth; $x'_i$ and $y'_i$ are the coordinate deviations from the center of the county to the mean center, respectively; $\delta_x$ and $\delta_y$ are the standard deviations along the $x$ and $y$ axes, respectively.

*3.5. Trends in Ecological-Economic Values*

In order to evaluate and prove the spatial distribution and intensity of ecological-economic values in Guangxi, the interannual rate of change of ecological-economic values was analyzed based on a linear regression equation to define the trend of ecological-

economic values based on the reference to related studies, with 10 km × 10 km grid as the research scale [59,61].

$$ECT = \frac{\sum\limits_{i=1}^{n} iESV_a - \frac{1}{n}\sum\limits_{i=1}^{n} i \sum\limits_{i=1}^{n} ESV_a}{\sum\limits_{i=1}^{n} i^2 - \frac{1}{n}\left(\sum\limits_{i=1}^{n} i\right)^2} \tag{7}$$

where: *i* is a specific year and *ESVa* is the ecological value or economic value of an individual grid.

## 4. Results

### 4.1. Temporal Changes in Ecological-Economic Values

From Figure 3, it can be seen that the ecological-economic values of Guangxi and each consolidation area develop in the opposite direction, showing that the ecological values fluctuate and decrease, and the economic values continue to increase. It can be seen that in order to promote socio-economic development, most regions sacrifice their ecological capital in exchange for economic capital. From the ecological value folding graph, the largest total ecological value of the five major consolidation areas is the North Mountain area in western Guangxi, and the smallest is the karst basin area in central Guangxi. The maximum value is about 2.5 times the minimum value. The economic value line graph shows that the economic value of the regions as a whole is increasing and reaches a maximum in 2020. This indicates that over time, regions have continued to increase their economic value based on their excellent ecological resources to meet their industrialization and urbanization development. Among them, the total economic value of the coastal hilly plain area is at the forefront, with about 3.5 times the smallest value of Guangxi's northwest mountain area.

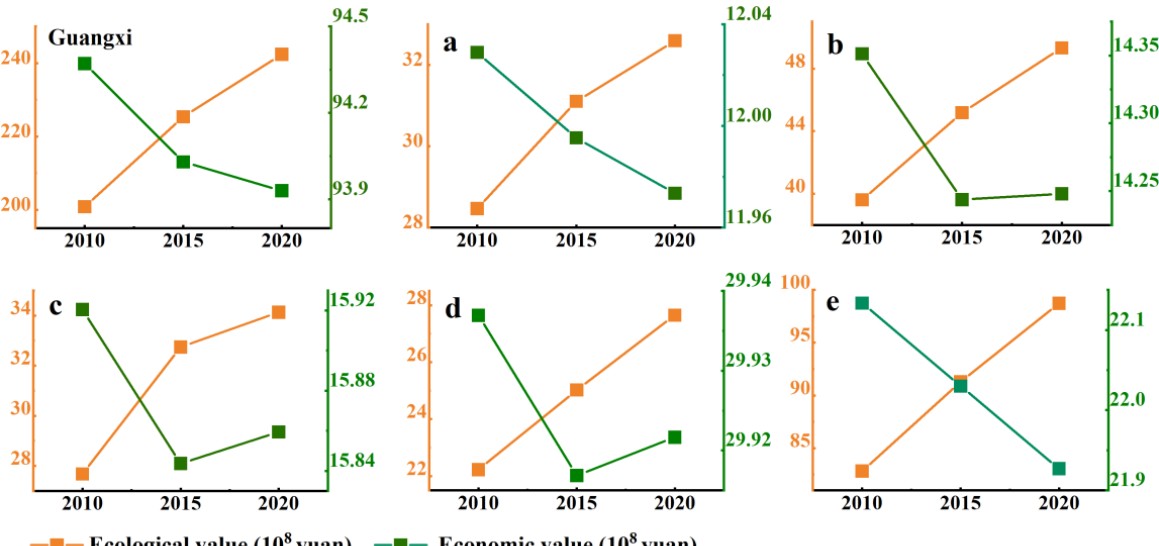

**Figure 3.** The trend of ecological-economic value changes in Guangxi and each consolidation area. (Note: (**a**) is the middle karst basin area of Guangxi, (**b**) is the southeast plain area of Guangxi, (**c**) is the northeast hilly mountain area of Guangxi, (**d**) is the northwest mountain area of Guangxi, (**e**) is the coastal hilly plain area).

### 4.2. Spatial Distribution of Eco-Economic Values

Based on ArcGIS 10.2 software, the spatial differences of ecological-economic values in Guangxi and each consolidation area are represented using a 10 km × 10 km grid scale, and classified into five grades of low, low, medium, high, and very high using the natural interruption point method. Figures 4 and 5 show that the spatial divergence of ecological-economic values in Guangxi is obvious. The ecological value shows the

distribution characteristics of "high around–low in the middle". The high-value areas of economic value are distributed in a pattern of "four vertical and two horizontal", while the low-value areas are distributed around them.

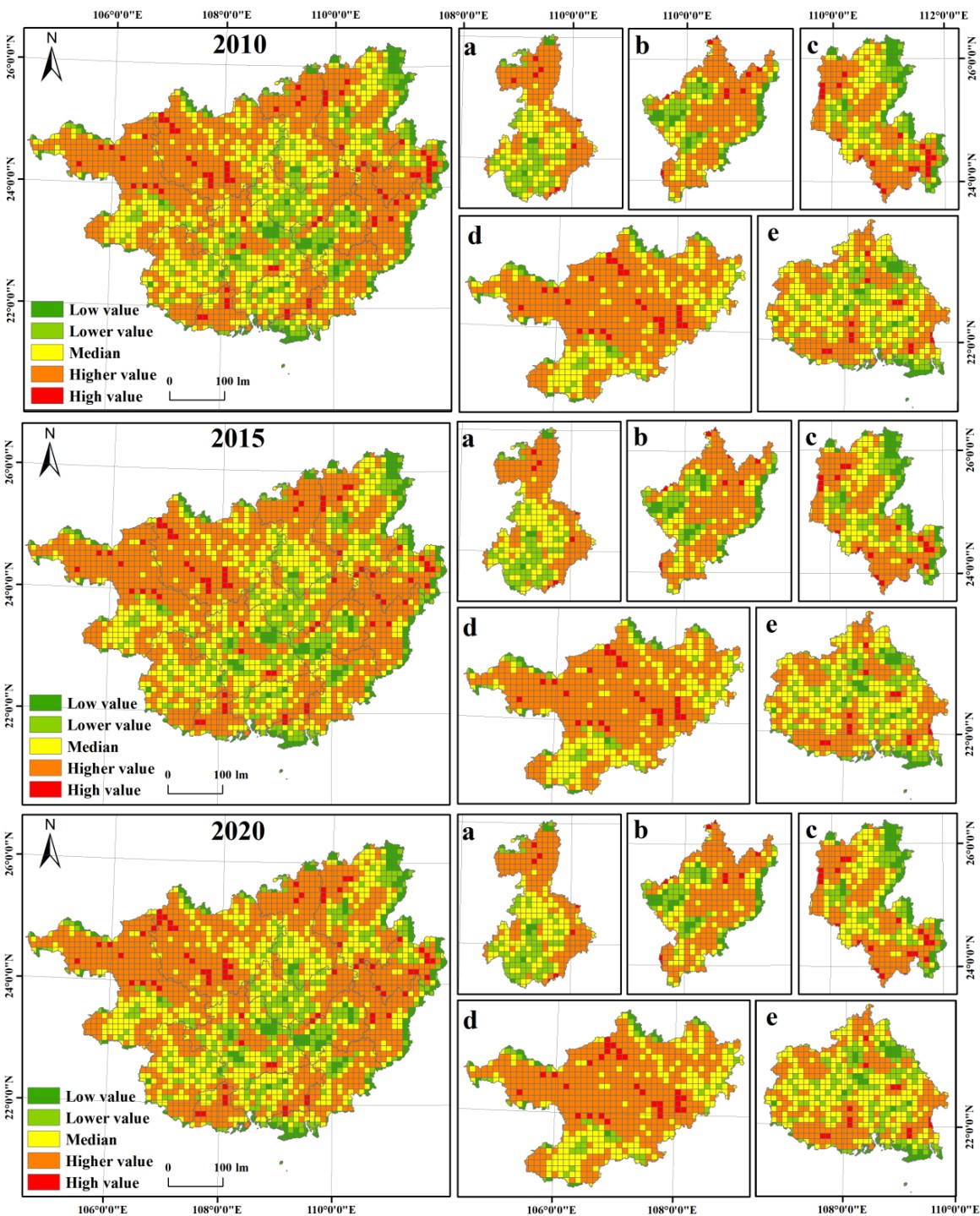

**Figure 4.** Ecological value grid distribution map of Guangxi and each consolidation area. (Note: (**a**) is the middle karst basin area of Guangxi, (**b**) is the southeast plain area of Guangxi, (**c**) is the northeast hilly mountain area of Guangxi, (**d**) is the northwest mountain area of Guangxi, (**e**) is the coastal hilly plain area).

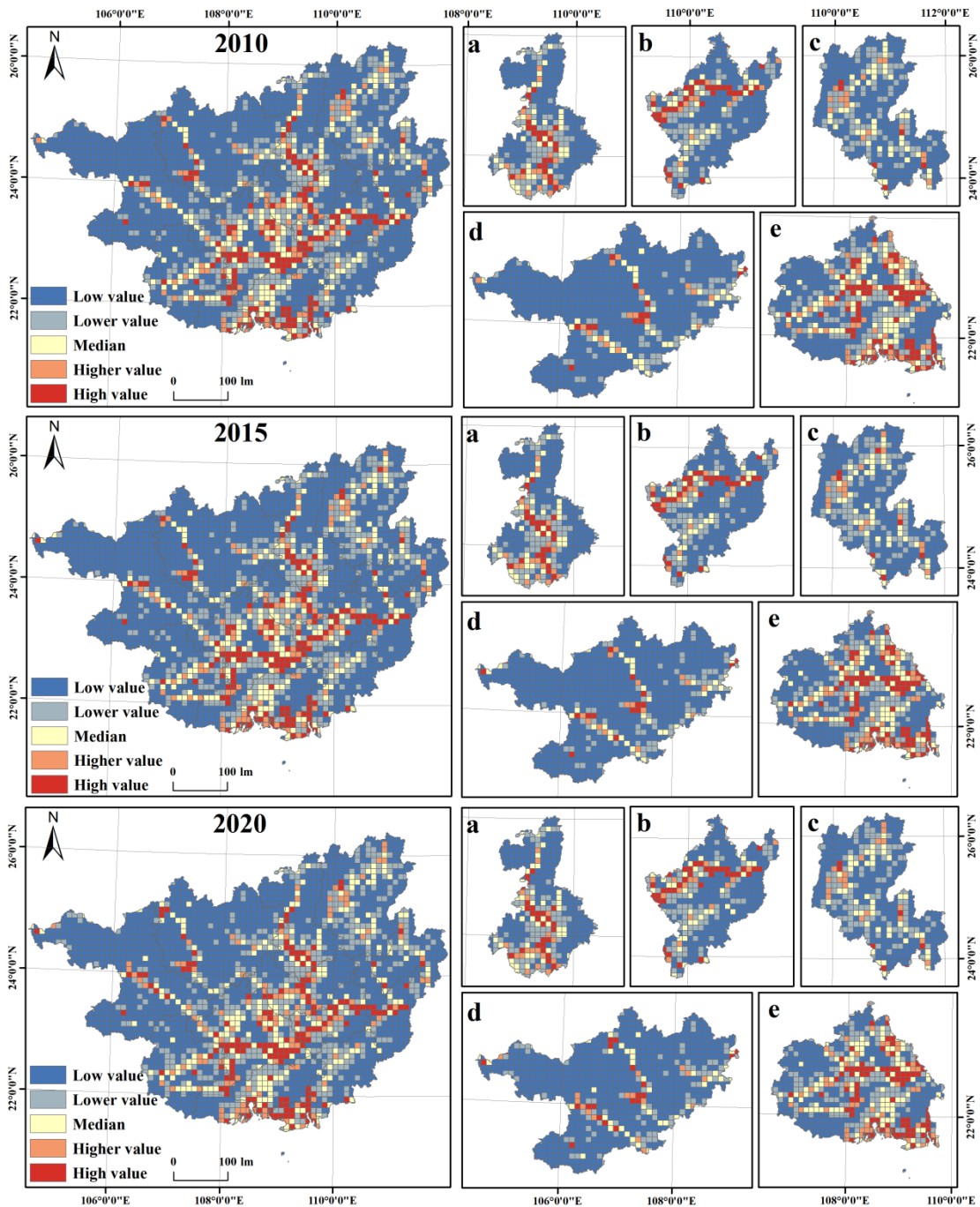

**Figure 5.** Economic value grid distribution map of Guangxi and each consolidation area. (Note: (**a**) is the middle karst basin area of Guangxi, (**b**) is the southeast plain area of Guangxi, (**c**) is the northeast hilly mountain area of Guangxi, (**d**) is the northwest mountain area of Guangxi, (**e**) is the coastal hilly plain area).

In the ecological value distribution map, it is obvious that the high-value areas are concentrated in the mountainous areas of northwest Guangxi and the northeastern part of the plain area of southeast Guangxi. The area has a large amount of woodland and grassland cover and is rich in forest and grass resources. The low-value areas are mainly distributed in the northeastern part of the hilly mountain area in northeastern Guangxi and the southwestern part of the karst basin area in central Guangxi, especially in the central part of the coastal hilly plain area. The coastal hilly plain area, as the economic zone of Beibu Gulf in Guangxi, shows an obvious trend of expansion of the low-value area over time.

The region serves as an important corridor for the new western land and sea corridor and the China-ASEAN Economic Free Trade Area, during which a large amount of ecological land was occupied for the promotion of urbanization and industrialization (Figure 4).

In contrast to the regional distribution of ecological values, in the economic value distribution map, the high-value areas are significantly clustered in the central part of the coastal hilly plain area and along the coast, while in other consolidation areas, they show a linear distribution in the core urban areas of the municipalities. This is because the coastal hilly plain area has excellent locational advantages, high population density, and a high level of economic development. The core urban areas of the municipalities in the other consolidation zones are the centers of social, economic, political, and cultural development. The low-value areas are mainly concentrated in the mountainous areas of northwest Guangxi. Due to the topography of the northwestern karst mountains, transportation accessibility is low, and public service facilities are not sound and are in old, marginal, and poor areas, so it is difficult to develop resources and transform ecological capital into economic capital (Figure 5).

### 4.3. Bivariate Spatial Autocorrelation of Ecological-Economic Values

Using the calculated value of ecological value as the first variable and economic value as the second variable, the bivariate spatial autocorrelation was used to explore the relationship between the two. As shown in Figure 6, Moran's I was negative for all groups and passed the significant level test of $p < 0.01$, indicating a significant negative correlation between ecological-economic values. That is, when the ecological value increases, the economic value all show a decreasing trend. As shown by the bivariate LISA clustering diagram of ecological-economic values, grids with high and low aggregation statuses dominated the study area, and a large number of them were distributed in the northwest mountain area of Guangxi (Figure 7). This indicates the presence of a contiguous grid of high ecological value in the region when the economic value it can provide is low. This may have resulted in a large number of low-low aggregation lattices mosaicked around the presence of high-low aggregation lattices, but overall there is a decreasing trend in the number of this aggregation type. The number of grids with low and high aggregation status is the second highest, and the "clusters" are distributed in the central city of each city. The grid of high and high aggregation states is sporadically distributed around the low and high aggregation states. And it shows an increasing trend with time, and the area has good potential for ecological-economic value transformation.

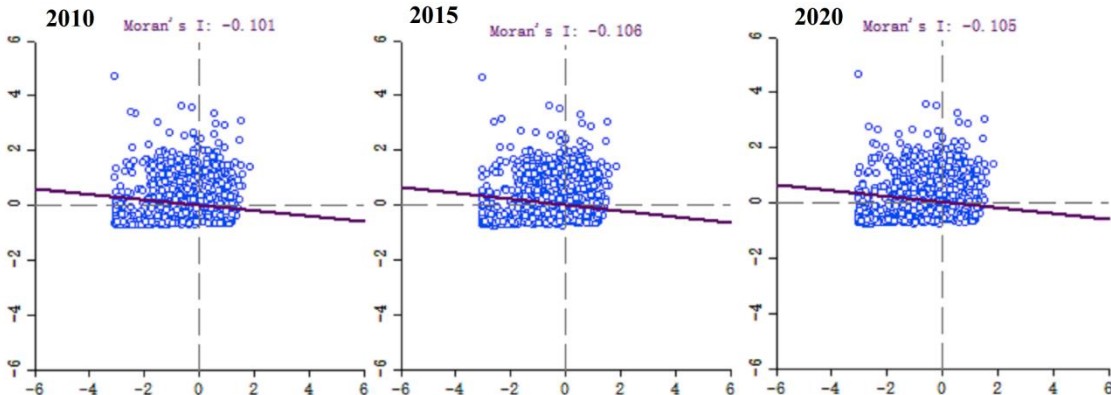

**Figure 6.** Ecological-economic value bivariate global spatial autocorrelation Moran index map.

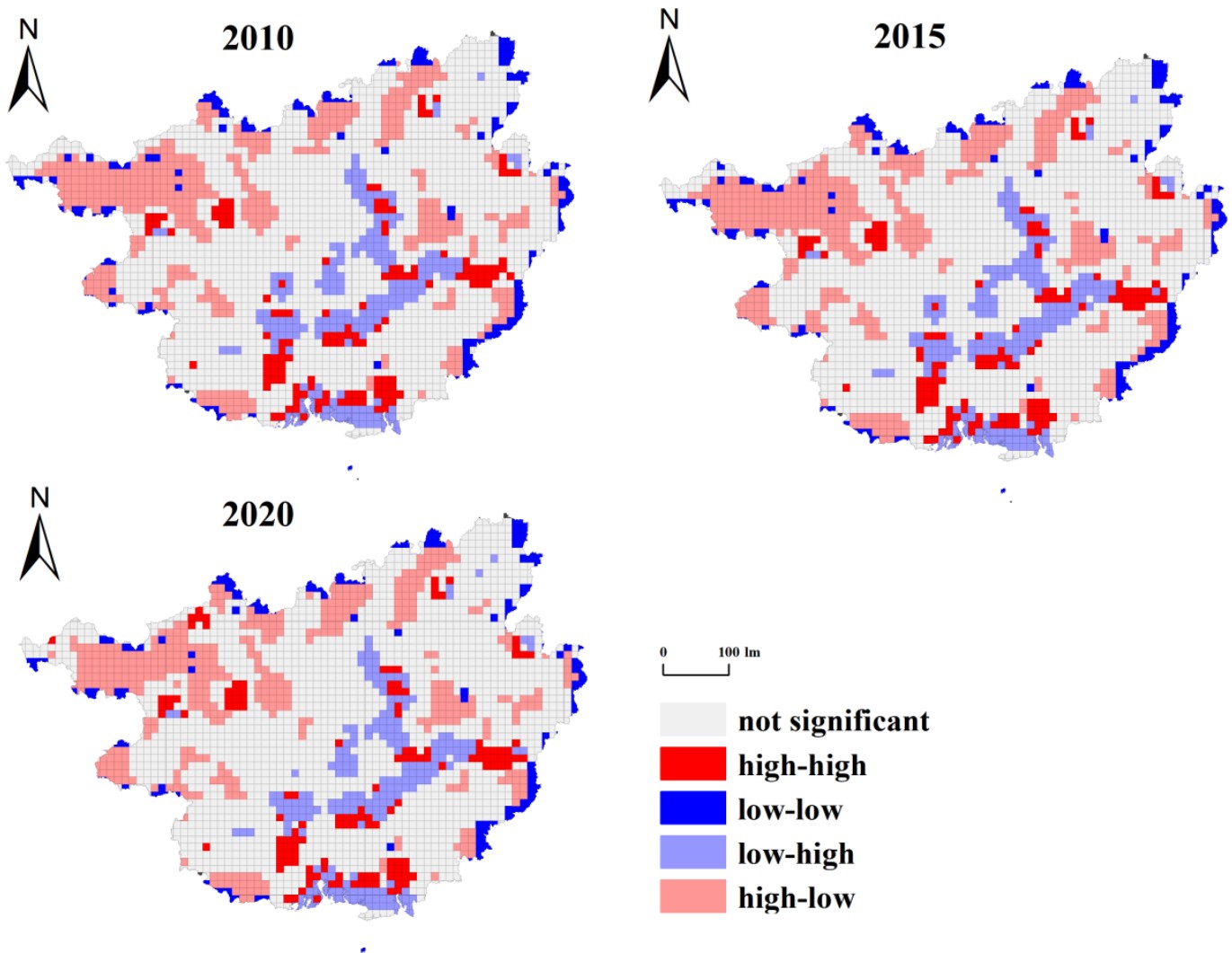

**Figure 7.** Bivariate local spatial autocorrelation distribution maps of ecological-economic values.

*4.4. Standard Deviation Ellipse Analysis of Ecological-Economic Values*

Standard deviation ellipses with center of gravity locations were plotted using ArcGIS 10.2 software to explore the overall characteristics of the eco-economic values and the state and direction of shifts occurring along the time series (Figure 8). From the expansion changes in different directions, the center of gravity of ecological-economic values shifted to the northwest overall during the study period, and the center of gravity locations all remained in the central karst basin area of Guangxi. This shows that the central karst basin area in Guangxi has good production-life-ecological functions and is an excellent ecological-economic system that promotes the transformation of ecological-economic values. In addition, the ellipse area of ecological value has been shrinking and the ellipse area of economic value has been continuously expanding. With the socio-economic development, the study area occupied a large amount of ecological land for promoting urbanization and industrialization during the period. The continuous promotion of urban construction and industrial parks and the continuous loss of ecological land means the oval area of ecological-economic value shows the opposite trend.

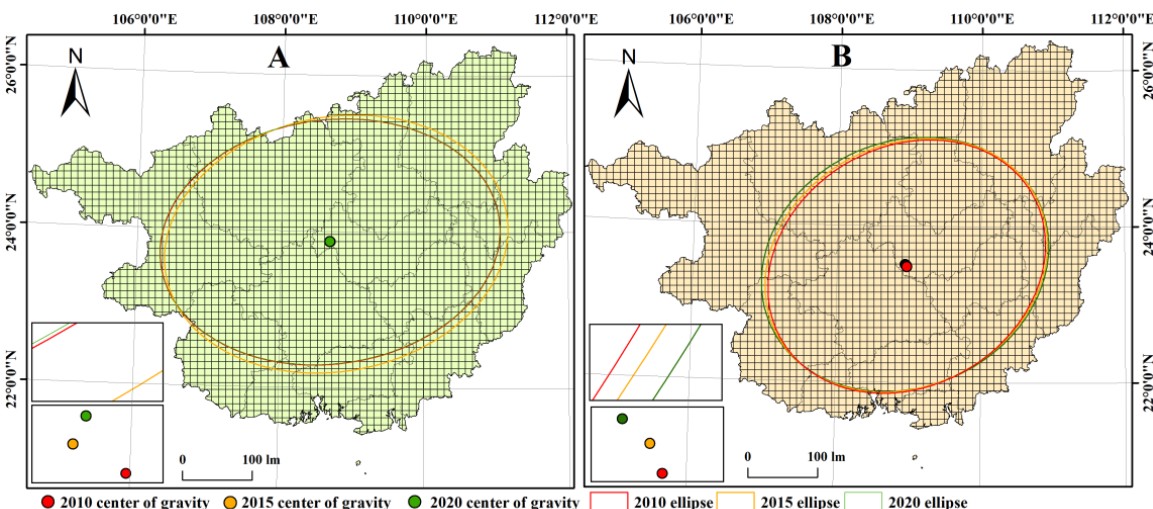

**Figure 8.** Elliptical variation of standard deviation of ecological-economic values. (Note: (**A**) is ecological value, (**B**) is economic value).

*4.5. Trends in Ecological-Economic Values before and after Land Consolidation*

To evaluate the spatial distribution and intensity of ecological-economic value changes in Guangxi, the trends of ecological-economic value changes in Guangxi were explored with a grid as the research unit based on three key time points before and after land consolidation in 2010, 2015, and 2020. The interlocking distribution of the grid of improved and deteriorated eco-economic values in Guangxi can be found in Figure 9. The grid of ecological value and economic value deterioration and improvement shows a more opposite distribution. The number of grids with deteriorating ecological value status dominates the region absolutely, at about 67%. The economic value is the opposite, with a grid number share of about 66% for the improved state. The trend of ecological-economic value change in each consolidation area is more consistent with Guangxi. Among them, the southeast plain area of Guangxi has the largest amount of ecological value deterioration, accounting for up to about 75%. The middle karst basin area in Guangxi has the highest number of economic value improvement grids, accounting for up to about 75% (Figure 10).

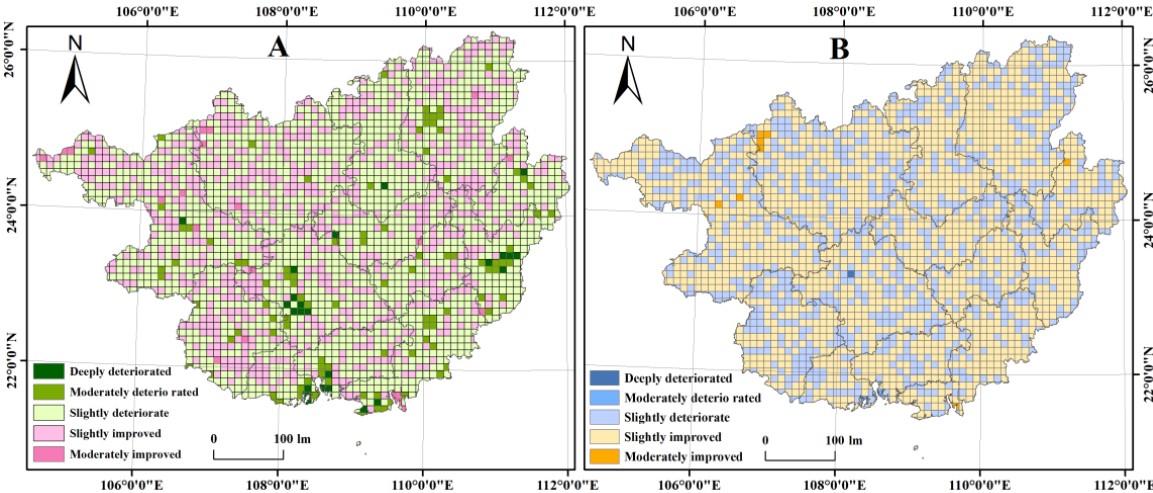

**Figure 9.** Map of spatial distribution of trends in ecological-economic value changes. (Note: (**A**) is ecological value, (**B**) is economic value).

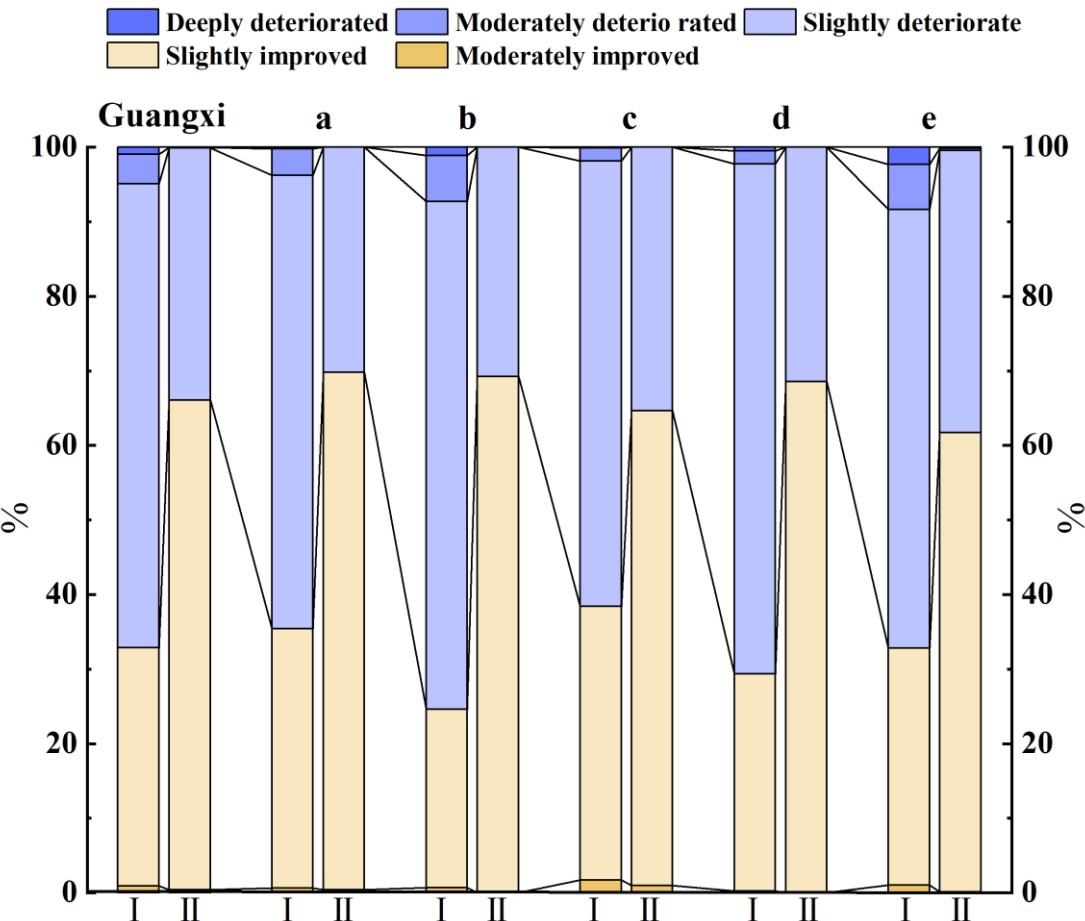

**Figure 10.** Percentage of ecological-economic value change types in Guangxi and each consolidation area. (Note: (**a**) is the middle karst basin area of Guangxi, (**b**) is the southeast plain area of Guangxi, (**c**) is the northeast hilly mountain area of Guangxi, (**d**) is the northwest mountain area of Guangxi, (**e**) is the coastal hilly plain area).

## 5. Discussion

### 5.1. Impact of Land Use Change on eco-Economic Value before and after Land Consolidation

Figure 11 clearly portrays the spatial distribution characteristics and quantitative change patterns of the intersectional transformation of land use types in each region. In terms of time, the total area of land use type change in Guangxi from 2010 to 2020 is about 7528.09 km$^2$. The conversion of forestland to cropland, cropland to forestland, and forestland to grassland are the largest land use conversion patterns by area in each region. Other scholars also found that outflow and inflow of forestland are the main types of land use conversion in Guangxi [62]. The various policies implemented by the government in recent years, such as returning farmland to forest and grass, have played a great role in keeping the ecological value of the region stable [63,64]. During the period of land consolidation, the social and economic development and urban and rural expansion in Guangxi have led to the continuous expansion of construction land, and its area has expanded mainly at the expense of arable land, forestland, and grassland.

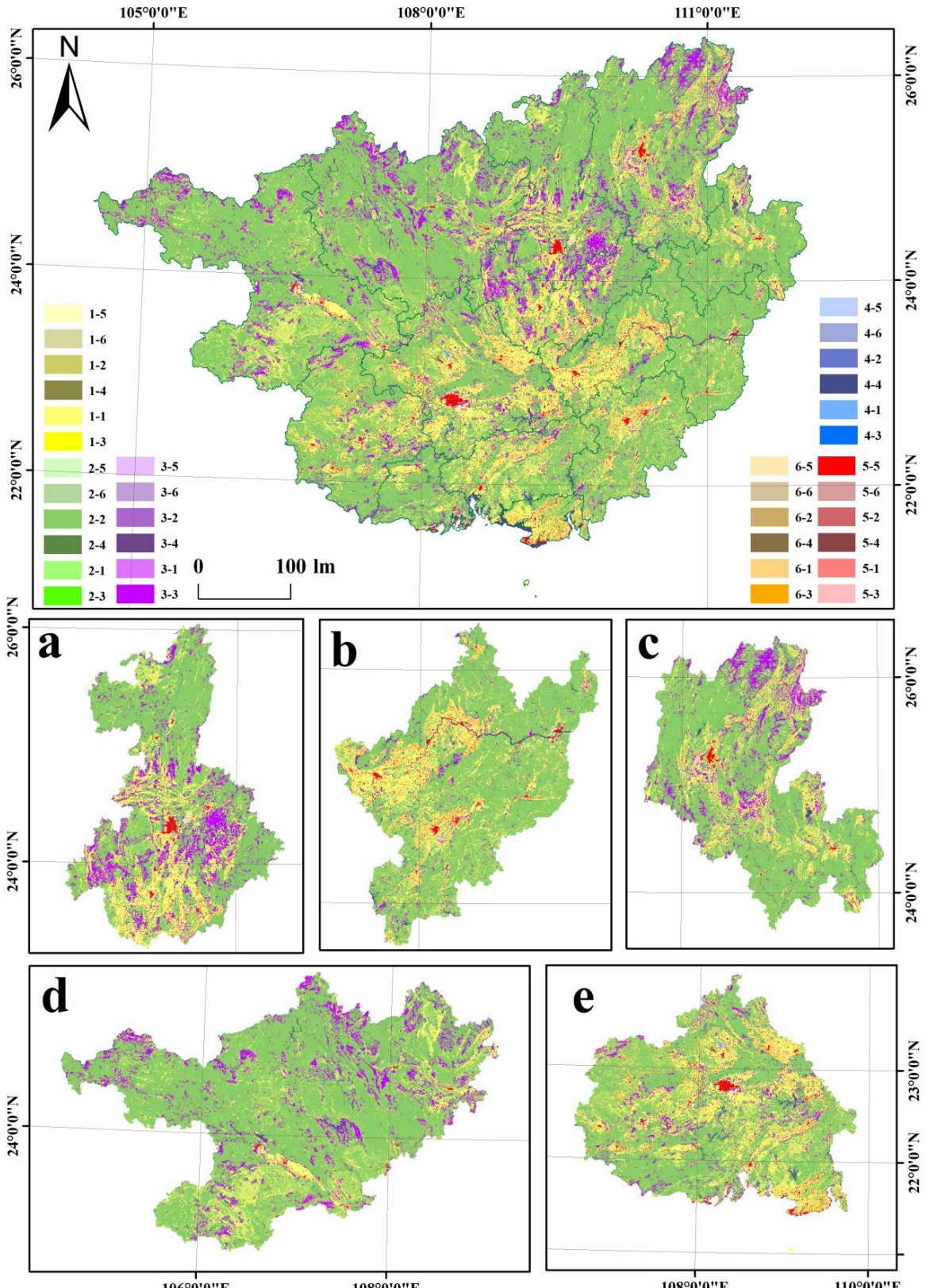

**Figure 11.** Cross distribution of land use in Guangxi and each consolidation area. (Note: (**a**) is the middle karst basin area of Guangxi, (**b**) is the southeast plain area of Guangxi, (**c**) is the northeast hilly mountain area of Guangxi, (**d**) is the northwest mountain area of Guangxi, (**e**) is the coastal hilly plain area).

The ecological value of Guangxi continued to decrease during the study period, with an average annual decrease of 0.23%. The economic value has been increasing, with an average annual increase of 9.9% (Figure 3). As can be seen from Tables 3 and 4, land use shifts in the study area resulted in changes in eco-economic values. The conversion of cropland, construction land, and grassland to forestland and the conversion of cropland to construction land had the most significant impact on the reduction in ecological value, with a contribution of 41.10%. Among them, the encroachment of forestland on cropland was the most significant in reducing the ecological value with a contribution of 21.33%. From 2010 to 2020, the decline in ecological value was relatively small, and the main reason for promoting the maintenance of stable ecological value was the shift from cropland and grassland to forestland, which contributed 27.51% to the enhancement of ecological value (Table 3). The various policies implemented by the government in recent years, such as returning farmland to forestland and grassland, have played a great role in keeping the ecological value of the region stable. Table 4 shows that the conversion of cropland and forestland to building land made the most significant contribution to the increase in economic value, with a contribution of 69.14%. Among them, the conversion of arable land into construction land made the most obvious contribution to increasing economic value, with a contribution rate of 46.06%.

**Table 3.** Effect of land consolidation before and after on the number of ecological values ($10^4$ yuan).

| 2010→2020 | 2010 | 2020 | Profit and Loss | Contribution Margin (%) |
|---|---|---|---|---|
| Grassland→Farmland | 56,516.88 | 38,258.62 | −18,258.25 | 0.65% |
| Grassland→Construction land | 35,322.37 | −4752.51 | −40,074.88 | 1.43% |
| Grassland→Forestland | 135,254.44 | 325,904.37 | 190,649.94 | 6.82% |
| Grassland→Water area | 11,026.28 | 47,298.48 | 36,272.21 | 1.30% |
| Grassland→Unused land | 106.50 | 12.68 | −93.81 | 0.00% |
| Farmland→Grassland | 34,844.50 | 51,473.43 | 16,628.93 | 0.59% |
| Farmland→Construction land | 165,096.16 | −32,814.01 | −197,910.18 | 7.08% |
| Farmland→Forestland | 226,076.18 | 804,715.34 | 578,639.16 | 20.69% |
| Farmland→Water area | 25,953.86 | 164,463.33 | 138,509.47 | 4.95% |
| Farmland→Unused land | 336.05 | 59.12 | −276.93 | 0.01% |
| construction land→Grassland | −537.63 | 3995.89 | 4533.53 | 0.16% |
| construction land→Farmland | −8805.38 | 44,302.23 | 53,107.61 | 1.90% |
| construction land→Forestland | −1680.37 | 30,093.36 | 31,773.73 | 1.14% |
| construction land→Water area | −805.11 | 25,668.53 | 26,473.65 | 0.95% |
| construction land→Unused land | −10.49 | 9.29 | 19.78 | 0.00% |
| Forestland→Grassland | 423,986.24 | 175,959.65 | −248,026.59 | 8.87% |
| Forestland→Farmland | 829,509.33 | 233,041.78 | −596,467.55 | 21.33% |
| Forestland→Construction land | 288,817.94 | −16,127.20 | −304,945.15 | 10.90% |
| Forestland→Water area | 100,600.47 | 179,093.42 | 78,492.96 | 2.81% |
| Forestland→Unused land | 2482.73 | 122.71 | −2360.03 | 0.08% |
| Water area→Grassland | 27,391.52 | 6385.54 | −21,005.98 | 0.75% |
| Water area→Farmland | 122,243.88 | 19,291.23 | −102,952.64 | 3.68% |
| Water area→Construction land | 53,566.34 | −1680.15 | −55,246.49 | 1.98% |
| Water area→Forestland | 107,956.40 | 60,641.34 | −47,315.07 | 1.69% |
| Water area→Unused land | 2682.50 | 74.47 | −2608.02 | 0.09% |
| Unused land→Grassland | 71.70 | 602.08 | 530.38 | 0.02% |
| Unused land→Farmland | 14.01 | 79.61 | 65.61 | 0.00% |
| Unused land→Construction land | 186.80 | −211.04 | −397.83 | 0.01% |
| Unused land→Forestland | 36.92 | 747.09 | 710.17 | 0.03% |
| Unused land→Water area | 61.59 | 2218.48 | 2156.89 | 0.08% |
| Total | 2,638,302.60 | 2,158,927.18 | −479,375.41 | 100.00% |

**Table 4.** Effect of land consolidation before and after on the quantity of economic value (104 yuan).

| 2010→2020 | 2010 | 2020 | Profit and Loss | Contribution Margin (%) |
|---|---|---|---|---|
| Grassland→Farmland | 167,427.41 | 156,118.78 | −11,308.63 | 0.02% |
| Grassland→Construction land | 104,640.11 | 4,597,457.77 | 4,492,817.66 | 6.66% |
| Grassland→Forestland | 400,682.09 | 16,369.27 | −384,312.82 | 0.57% |
| Grassland→Water area | 32,664.60 | 65,118.84 | 32,454.24 | 0.05% |
| Grassland→Unused land | 315.49 | 0.00 | −315.49 | 0.00% |
| Farmland→Grassland | 142,187.07 | 152,486.54 | 10,299.47 | 0.02% |
| Farmland→Construction land | 673,694.17 | 31,743,422.29 | 31,069,728.12 | 46.04% |
| Farmland→Forestland | 922,530.23 | 40,418.62 | −882,111.62 | 1.31% |
| Farmland→Water area | 105,907.76 | 226,427.17 | 120,519.41 | 0.18% |
| Farmland→Unused land | 1371.28 | 0.00 | −1371.28 | 0.00% |
| construction land→Grassland | 520,093.67 | 11,837.56 | −508,256.11 | 0.75% |
| construction land→Farmland | 8,518,092.54 | 180,780.42 | −8,337,312.11 | 12.35% |
| construction land→Forestland | 1,625,548.52 | 1511.51 | −1,624,037.02 | 2.41% |
| construction land→Water area | 778,844.43 | 35,339.51 | −743,504.92 | 1.10% |
| construction land→Unused land | 10,150.30 | 0.00 | −10,150.30 | 0.02% |
| Forestland→Grassland | 21,295.65 | 521,268.53 | 499,972.88 | 0.74% |
| Forestland→Farmland | 41,663.95 | 950,954.21 | 909,290.26 | 1.35% |
| Forestland→Construction land | 14,506.52 | 15,601,036.51 | 15,586,529.99 | 23.10% |
| Forestland→Water area | 5052.88 | 246,569.35 | 241,516.47 | 0.36% |
| Forestland→Unused land | 124.70 | 0.00 | −124.70 | 0.00% |
| Water area→Grassland | 37,711.66 | 18,916.74 | −18,794.92 | 0.03% |
| Water area→Farmland | 168,300.95 | 78,720.13 | −89,580.82 | 0.13% |
| Water area→Construction land | 73,748.20 | 1,625,330.23 | 1,551,582.04 | 2.30% |
| Water area→Forestland | 148,630.47 | 3045.85 | −145,584.63 | 0.22% |
| Water area→Unused land | 3693.17 | 0.00 | −3693.17 | 0.01% |
| Unused land→Grassland | 0.00 | 1783.61 | 1783.61 | 0.00% |
| Unused land→Farmland | 0.00 | 324.87 | 324.87 | 0.00% |
| Unused land→Construction land | 0.00 | 204,151.98 | 204,151.98 | 0.30% |
| Unused land→Forestland | 0.00 | 37.52 | 37.52 | 0.00% |
| Unused land→Water area | 0.00 | 3054.32 | 3054.32 | 0.00% |
| Total | 14,518,877.82 | 56,482,482.13 | 41,963,604.31 | 100.00% |

## 5.2. Optimization of Land Consolidation Zoning under Ecological-Economic Value Matching

The ecological-economic values of each consolidation area obtained separately were standardized by z-score, and the standardized results were matched by dividing quadrants for analysis [65–67]. The *x*-axis represents the result of economic value standardization and the y-axis represents the result of ecological value standardization. These four quadrant types are low ecological-economic value imbalance (quadrant I), ecological value imbalance (quadrant II), economic value imbalance (quadrant III), and ecological-economic value coordinated development (quadrant IV) (Figure 12).

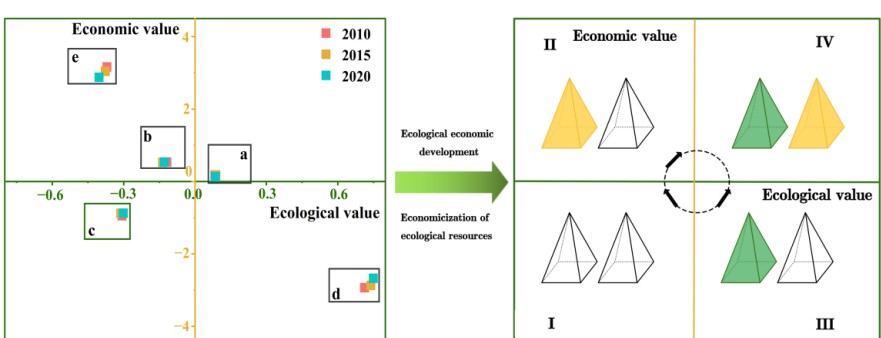

**Figure 12.** Spatial zoning optimization map of land consolidation with matching ecological-economic values. (Note: a is the middle karst basin area of Guangxi, b is the southeast plain area of Guangxi, c is the northeast hilly mountain area of Guangxi, d is the northwest mountain area of Guangxi, e is the coastal hilly plain area. I indicates the type of low ecological-economic value imbalance, II indicates the type of ecological value imbalance, III indicates the type of economic value imbalance, and IV indicates the type of coordinated ecological-economic value development).

The Guangxi central karst basin region (ecological and economic value coordinated development type). The geographical conditions in the area are good, with superior natural conditions such as water, soil, light, and heat, and the ecological background is good. The forest cover is high, at about 59%. The arable land resources are rich, and the arable land area accounts for about 23% of the total area of the consolidation area. At the same time, Liuzhou is an industrial manufacturing production base and a transportation hub in the southwest part of the region. Laibin City is an emerging modern industrial city. The area has a high population density, well-developed agricultural, forestry, and industrial industries, high levels of socio-economic development and urbanization [68], and is a high supply area of ecological-economic value. These findings are consistent with the previous research of Xie et al. [68]. In addition, the location of the center of gravity of ecological-economic values was maintained in the middle karst basin area of Guangxi before and after land consolidation, which also demonstrates the excellent potential of this consolidation area to promote the transformation of ecological-economic values (Figure 8). The key to the transformation of the "Two Mountains" theory is to promote the transformation of ecological advantages to economic output, that is, the realization of the value of ecological products [69]. The district in the process of land consolidation ecological transformation can continue to broaden the path of transformation of the "Two Mountains" by clarifying land property rights, clarifying responsibility positioning, and building a consolidation transformation platform to support the realization of the value of ecological products [70,71].

The southeast Guangxi plain area (ecological value imbalance type). The district has reduced forest quality and relatively few natural resources. The supply capacity of ecosystem services can barely meet its own demand, and the supply of ecological value is low. In order to optimize the industrial structure and promote urbanization, a large amount of arable land and forestland was converted into construction land in the land use transfer from 2010 to 2020, and the total area transferred was up to 370.12 km$^2$ (Figure 11). In addition, the southeastern plain area of Guangxi had the largest amount of ecological value deterioration, accounting for up to about 75% (Figure 10). Therefore, the direction of land consolidation in this area should promote the evolution of the ecological value imbalance area II to the ecological-economic value coordination equilibrium state area IV. First, it should strengthen natural vegetation restoration, improve forest cover and protect the integrity of the ecosystem [72]. Second, to carry out sloping soil erosion control, landslide mudslide disaster prevention and control, closed mountain forestry, and land improvement projects [73]. Third, to adjust the industrial structure and production layout of the region [74]. Development should focus on the agricultural industry based on the Xunyu Plain, supported by the Yu River along with the Xunjiang River and its tributaries surrounding reservoirs. Large-scale, high-standard farmland should be built to ensure the safety of food production in Guangxi. Among them, Yulin City should be combined with modern special agricultural demonstration areas and the construction of high-standard farmland with local characteristics.

The hilly mountain area in northeastern Guangxi (low ecological-economic value imbalance type). The area has limited natural resources, relatively low level of industrial structure optimization and urbanization, and slow economic development, which is an imbalance type with low ecological-economic value. This finding is more consistent with the previous study by Xie et al. [68]. Guilin and Hezhou are positioned in the functional area planning as new industrial and high-tech industrial bases, building international tourist cities and ecological cities. With the increasing demand for industrial and tourist land and frequent human activity, the area's natural ecosystems have been damaged by various human activities, the quality of the forests has decreased, and biodiversity has been reduced. Mining and abandoned land have caused local environmental pollution and ecological damage, and a large amount of ecological land has been occupied. In addition, in the local spatial autocorrelation distribution map of ecological-economic value bivariate, it can be found that the grid of the district is mainly insignificant state and low-low aggregation (Figure 7). For this reason, the land consolidation direction of this consolidation area

should advance from state I to state II or state III, and finally evolve to state IV. First, the region should arrange urban and rural construction land in an integrated manner, adhere to land conservation and intensive development, strengthen the finishing of rural settlements and land reclamation of abandoned mining areas, and continuously improve the level of intensive land use [75]. Second, the region should focus on ecological improvement, pay attention to ecological restoration and ecosystem service function enhancement, continuously improve urban green infrastructure and corridor landscape in urban areas, increase the area of parks and recreational green space, and improve vegetation coverage [72]. Third, the region should adopt comprehensive measures to control soil erosion [73], adjust industrial structure and production layout [74], expand ecological land space (such as recreational green space and parks) [76], and weigh ecological civilization construction and economic development comprehensively.

The northwest mountain region of Guangxi (economic value imbalance type). The ecological background of the region is good, with high vegetation cover. It is not only an important water connotation area and an important biodiversity protection area in Guangxi but also an important ecological barrier in the upper and middle reaches of the Xijiang River Basin. However, the area has serious rock desertification, frequent natural disasters and low land utilization [77]. The economic base is also relatively weak, and the population density is low; in addition, it is a border area where ethnic minorities gather, and most of the poor counties in Guangxi are located here [77]. Since human activities have less impact on the regional ecology, they are in a state of economic value imbalance. This can be seen from Figure 3, which demonstrates that the five major consolidation areas with the largest total ecological value and the smallest total economic value are the northwestern mountain areas of Guangxi. Similarly, a patchy distribution of ecological-economic value high and low aggregation states can be found here in Figure 7. The direction of land consolidation in the area should advance from state III to state IV. First of all, it focuses on the conservation and maintenance of ecosystem service functions, strengthening the protection and restoration of water-conserving forests, construction and management of forest nature reserves, and building biological corridors to strengthen habitat connectivity while improving forest quality and preventing large-scale regional development and rock desertification. Second, to carry out ecological construction land consolidation and precise poverty alleviation land consolidation, high-standard farmland is mainly constructed on dry slopes [78], combined with non-ferrous metal and bauxite mining production in the two cities to promote green mine land reclamation activities.

The coastal hilly plain area (ecological value imbalance type). The region has Nanning as the core, Nanning to Binhai as the main axis, and a comprehensive transport channel as the link to the Beibu Gulf city cluster. The region is flat, with high population density and gradually increasing urbanization, and is in an overall state of ecological value imbalance. With the support of policies such as the 2007 National Western Development Plan, the 2008 Guangxi Beibu Gulf Economic Zone Development Plan, the 2010 Guangxi Beibu Gulf Economic Zone Town Cluster Planning Outline, and the 2017 Beibu Gulf City Cluster Development Plan, the region has been vigorously developing its sea-side and port-side industries [79]. Coordinated construction has occurred in industrial development, transportation and logistics, infrastructure, culture, and education, etc. Its unique coastal ecological and locational advantages promote the cross-transfer of various types of land. The largest area of land use transfer was up to 2352.28 km$^2$ (Figure 11). The largest total economic value of the five consolidation areas is in the coastal hilly plain land consolidation area (Figure 3). Therefore, the direction of land consolidation in this area should promote its evolution from state II to state IV. First of all, the area should focus on ecological improvement and pay attention to ecological restoration and ecosystem service function enhancement. Secondly, we should insist on land conservation and intensive development, continuously improve urban green infrastructure and corridor landscape in urban areas, increase the area of parks and recreational green spaces, improve vegetation coverage,

expand ecological land space (such as recreational green spaces and parks), and weigh ecological civilization construction and economic development comprehensively [54].

### 5.3. Summary of the Effects of Land Use Change and Landform Type on Eco-Economic Values

Compared with existing studies [80–84], the gains and losses of ecological-economic values in consolidation areas with different land use changes and different landform types differed significantly. Land use change is a determinant of the ecological-economic value gain and loss in land consolidation areas. The results of the study show that an increase or decrease in ecological land types such as forestland directly affects the gain or loss of regional ecological value. This is consistent with the findings of Gu et al. [80] as well as Guo [81] and Zhang [82]. In addition, the conversion of arable land to building land has the most significant effect on adding economic value. The main reason for this is that the value coefficients of ecological services are clearly different for different land use types. For ecological value, the coefficient of ecological benefit value is higher for forestland, while the coefficient of economic benefit for construction land is the maximum in the accounting of economic value. Therefore, when implementing land consolidation, it is necessary to strengthen the protection of ecological land while giving full play to the economic benefits of construction land. Geomorphological conditions are also an important factor affecting the ecological-economic value gain and loss of each consolidation area. In this study, it was found that the overall ranking of the ecological-economic value of each landform improvement area was: plain > mountain > basin. This is consistent with the findings of Jiang et al. [82] and Yu et al. [84]. Due to the topographic factors, the land reclamation area of the plain landscape has good ecological-production-living functions and is concentrated on the area adjustment of various ecological function land types. In mountainous areas and basins, land consolidation not only adjusts the amount of ecologically functional land types but also improves the ecological and environmental elements of the project area through the projects and technologies used [40].

### 5.4. Limitations and Future Development

Existing studies have focused on the ways in which land consolidation policies promote sustainable rural development [85,86], from improving farmland infrastructure [87], increasing the productive potential of farmland [88], promoting human-land system coupling [89], and integrating land consolidation with industrial development. Although increasing regional economic value can improve regional development to a certain extent [78], economic value does not fully reflect the effectiveness of regional land consolidation, and the key goal of optimizing land consolidation areas is to solve the problem of balanced ecological-economic value. Unlike existing studies, this study provides an innovative tool for understanding the optimization of land consolidation zones based on the matching state of ecological and economic values and provides a theoretical basis for guiding land consolidation. However, unlike existing studies, this study innovatively provides an understanding tool for land consolidation zoning optimization based on the matching state of ecological and economic values, and provides a theoretical and practical basis for guiding how to realize the two-way transformation of ecological-economic values in each consolidation zone.

The methods for measuring economic value are now more mature. However, quantifying ecological values requires a large amount of basic data, which is a technical difficulty as well as a hot spot and focus of research in ecology and ecological economics. The most-used methods are the value equivalent method, market comparison method, cost substitution method, etc. This study calculates the value of ecosystem services based on the ecosystem service value equivalent factor proposed by Xie et al. [56–58]. There are still some controversies in the conceptual and practical aspects of its operation, and there is no broad consensus on how the quality of ecosystem services obtained from the assessment should be converted into a value quantity. Therefore, this study only reflects the ecological-economic value gains and losses due to the changes in the area of each category in the project area before

and after the implementation of the land consolidation project. However, there is a lack of consideration for changes in eco-economic values caused by changes in the quality of various localities and other hidden aspects such as construction. Future research should first focus on the characterization and measurement of ecological wealth, and construct a scientific measurement system of comprehensive indicators and a comprehensive driving model to account for various benefits, such as economic, social, and ecological benefits. Second, the driving process and driving mechanism of ecological-economic value from the perspective of land consolidation need to be explored.

## 6. Conclusions

This study takes land consolidation as the research perspective and explores the characteristics of ecological-economic value evolution and change intensity before and after land consolidation in Guangxi and different consolidation areas. The main results are as follows. (1) The ecological-economic values of Guangxi and the consolidation areas in the opposite direction demonstrated a fluctuating decrease in ecological value and a continuous increase in economic value. There was a significant negative correlation between the two. (2) From the changes in the expansion of ecological-economic values in different directions, the center of gravity as a whole moved to the northwest, and the center of gravity locations all remain in the middle karst basin area of Guangxi. (3) Among them, the ellipse area of ecological value keeps shrinking and the ellipse area of economic value keeps expanding. The ecological-economic value of each consolidation area was standardized by z-score and divided into quadrants for matching analysis. Their states are ecological-economic value coordinated development type (central karst basin area of Guangxi), ecological value imbalance type (southeast plain area and coastal hilly plain area of Guangxi), economic value imbalance type (northwest mountain area of Guangxi), and ecological-economic value low imbalance type (northeast hilly mountain area of Guangxi), respectively. Therefore, while promoting regional economic and social benefits, land consolidation can also have multiple impacts on the ecological benefits generated by natural ecosystems. The "Two Mountains" concept of "green water and green mountains are gold and silver mountains" plays an important value-oriented role in the ecological transformation of land consolidation. In order to better implement the "Two Mountains" concept, land consolidation should respond positively to the requirements of land use and economic and social transformation development, and pay attention to natural conditions and differences in local demand to realize the synergistic development of ecological and economic values and promote the comprehensive revitalization of economy, society, and ecology.

**Author Contributions:** Conceptualization, G.L. and S.H.; methodology, L.Z.; software, Z.Z.; validation, L.Z., G.L. and Z.Z.; formal analysis, B.H.; investigation, S.H.; resources, B.H.; data curation, L.Z.; writing—original draft preparation, L.Z.; writing—review and editing, L.Z.; visualization, L.Z.; supervision, Z.Z.; project administration, B.H.; funding acquisition, B.H. All authors have read and agreed to the published version of the manuscript.

**Funding:** This research was funded by the National Natural Science Foundation of China Projects grant number (No. 41930537 and No. 42071135). And this research was funded by the Guangxi science and technology base and talent special grant number (No. Gui AD19110142).

**Data Availability Statement:** The new data created in this study are available on request.

**Conflicts of Interest:** The authors declare no conflict of interest.

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
