# Peer review of "Comprehensive Evaluation of Ecological-Economic Value of Guangxi Based on Land Consolidation"

_land, doi:10.3390/land12040759_

Round 1

Reviewer 1 Report

The manuscript deals with the interesting and current topic of the Comprehensive evaluation of Guangxi based on land consolidation in relation of ecological-economic value. Overall the paper generate reasonable interest not only for scientific readers but also for general public. 

There seems a gap between data interpretation and its presentation though both attains high merit. Interpretation of data should be improved in the context of the work and there is a dire need of integrated interpretation in order to support the stated objectives.

In the introduction, Authors must update the literature review with some recent contributions. Furthermore, since the evaluation method that will be used is introduced, it is advisable to support the choice with biblio references that justify its use in the application.

Authors should also try to compare the results obtained in this study with the values obtained in similar studies in region or in other developing countries of the world.

Authors must improve the conclusions section and discuss/recommend some policies that local authorities could implement in order to improve the quality of ecological-economic value.

 The layout of the article is correct.

I enjoyed reading the article, it is structured clearly and legibly.

Author Response

Many thanks to the two experts for their valuable suggestions that have benefited this article! I have revised and responded to the article one by one according to the relevant suggestions.

Response to Reviewer 1

The manuscript deals with the interesting and current topic of the Comprehensive evaluation of Guangxi based on land consolidation in relation of ecological-economic value. Overall the paper generate reasonable interest not only for scientific readers but also for general public. 

1.There seems a gap between data interpretation and its presentation though both attains high merit. Interpretation of data should be improved in the context of the work and there is a dire need of integrated interpretation in order to support the stated objectives.

Response: Regarding the data interpretation, firstly, it has been briefly introduced in the 2.2. data source section; secondly, in the 4. result analysis section, the reasons for the important data interpretation have been analyzed; finally, in the 5.2. land remediation zoning optimization under ecological-economic value matching section, the ecological-economic values of the five major remediation areas have also been discussed in depth in conjunction with the 4. result analysis section.

2.In the introduction, Authors must update the literature review with some recent contributions. Furthermore, since the evaluation method that will be used is introduced, it is advisable to support the choice with biblio references that justify its use in the application.

Response: In the 1. Introduction section, older literature has been replaced with more recent references. In addition, relevant references have been cited to support the study in the 3. Research methods section.

3.Authors should also try to compare the results obtained in this study with the values obtained in similar studies in region or in other developing countries of the world.

Response: In the two discussion sections 5.1. and 5.2. some of these results analyses have been compared with similar studies in this study area.

4.Authors must improve the conclusions section and discuss/recommend some policies that local authorities could implement in order to improve the quality of ecological-economic value. The layout of the article is correct.

Response: The 6. Conclusion section has been improved and concludes with a recommendation of land reclamation policies that should be implemented in the study area to achieve synergistic development of ecological-economic values and contribute to comprehensive economic, social and ecological revitalization.

I enjoyed reading the article, it is structured clearly and legibly.

Reviewer 2 Report

Dear Authors,

The title of the study “Comprehensive evaluation of ecological-economic value of Guangxi based on land consolidation” corresponds to its content.

Keywords:  Land consolidation; Ecological-economic values; comprehensive evaluation; Guangxi are correct.

1.      The total value of work is a valuable contribution but references take only 64 publications are cited in the entire article. Literature research well started, but not enough publications. It is proposed to add the following articles that contain new research in this area, for example:

·         Basista, I.; Balawejder, M. 2020. Assessment of selected land consolidation in south-eastern Poland. Land Use Policy 2020, 99, 105033. https://doi.org/10.1016/j.landusepol.2020.105033

·         CienciaÅ‚a, A.; Sobura, S.; Sobolewska-Mikulska, K. Optimising Land Consolidation by Implementing UAV Technology. Sustainability 2022, 14, 4412. https://doi.org/10.3390/su14084412

·         Ertunç, E.; Muchová, Z.; Tomi´c, H.; Janus, J. Legal, Proceduraland Social Aspects of Land Valuation in Land Consolidation: A Comparative Study for Selected Central and Eastern Europe Countries and Turkey. Land 2022, 11, 636. https://doi.org/10.3390/land11050636

2. Similarly, the discussion or conclusion should refer to research conducted in this field in other countries and cited in this publication. Please complete this and the article will be a valuable scientific contribution.

3. There is a discussion section in this study, please complete it. Support the obtained results with research from other countries. Complete the citations - there are very few of them in this work. This will bring valuable scientific input to this article.

4. References do not comply with the requirements of the MDPI publication. You should [1] instead (Ruishi et al., 2019). Please correct it.

5. “2.1. Study area” description of this chapter is in bold type and should be in normal type. Justify it. Please correct it.

6. Figure 1 – a.china maybe is correct a.China ? Please improve the figure 1.

It should be noted that the whole of the study is cognitive and contains important scientific elements. The article was written at a good academic level. In relation to the above, I express the opinion that the work submitted for review should be published in its entirety after taking into account the comments of the reviewer but not require a review again.

Author Response

Many thanks to the two experts for their valuable suggestions that have benefited this article! I have revised and responded to the article one by one according to the relevant suggestions.

Response to Reviewer 2

1.The total value of work is a valuable contribution but references take only 64 publications are cited in the entire article. Literature research well started, but not enough publications. It is proposed to add the following articles that contain new research in this area, for example:

Basista, I.; Balawejder, M. 2020. Assessment of selected land consolidation in south-eastern Poland. Land Use Policy 2020, 99, 105033. https://doi.org/10.1016/j.landusepol.2020.105033

CienciaÅ‚a, A.; Sobura, S.; Sobolewska-Mikulska, K. Optimising Land Consolidation by Implementing UAV Technology. Sustainability 2022, 14, 4412. https://doi.org/10.3390/su14084412

Ertunç, E.; Muchová, Z.; Tomi´c, H.; Janus, J. Legal, Proceduraland Social Aspects of Land Valuation in Land Consolidation: A Comparative Study for Selected Central and Eastern Europe Countries and Turkey. Land 2022, 11, 636. https://doi.org/10.3390/land11050636

Response: The above article has been added, and other recent articles related to this study have been added.

2.Similarly, the discussion or conclusion should refer to research conducted in this field in other countries and cited in this publication. Please complete this and the article will be a valuable scientific contribution.

Response: Relevant citations have been completed for the 5. discussion section.

3.There is a discussion section in this study, please complete it. Support the obtained results with research from other countries. Complete the citations - there are very few of them in this work. This will bring valuable scientific input to this article.

Response: Firstly, some of these results analyses have been compared with similar studies in this study area in the two discussion sections 5.1. and 5.2. Secondly, relevant citations have been completed for the 5. discussion section.

4.References do not comply with the requirements of the MDPI publication. You should [1] instead (Ruishi et al., 2019). Please correct it.

Response: The reference citation format has been revised for the full text.

5.“2.1. Study area” description of this chapter is in bold type and should be in normal type. Justify it. Please correct it.

Response: The font formatting has been modified.

6.Figure 1 – a.china maybe is correct a.China ? Please improve the figure 1.

Response: Figure 1 has been improved, and "china" in Figure 1-a has been changed to "China".

It should be noted that the whole of the study is cognitive and contains important scientific elements. The article was written at a good academic level. In relation to the above, I express the opinion that the work submitted for review should be published in its entirety after taking into account the comments of the reviewer but not require a review again.
